# Transient domain boundary drives ultrafast magnetisation reversal

Martin Hennecke [1] ✉, Daniel Schick [1] ✉, Themistoklis P. H. Sidiropoulos [1], Jun-Xiao Lin [2], Zongxia Guo[2], Grégory Malinowski [2], Maximilian Mattern [1], Lutz Ehrentraut[1], Martin Schmidbauer [3], Matthias Schnuerer[1], Clemens von Korff Schmising[1], Stéphane Mangin [2], Michel Hehn [2] & Stefan Eisebitt [1,4]

Light-induced magnetisation switching is one of the most intriguing and promising areas where an ultrafast phenomenon can be utilised in technological applications. So far, experiment and theory have considered the origin of all-optical helicity-independent magnetisation switching (AO-HIS) in individual magnetic films only as a microscopically local, thermally-driven process of angular momentum transfer between different subsystems. Here, we demonstrate that this local picture is insufficient and that AO-HIS must also be regarded as a spatially inhomogeneous process along the depth within a few-nanometre thin magnetic layer. Two regions of opposite magnetisation directions are observed, separated by a highly mobile boundary, which propagates along the depth of a 9.4 nm thin $Gd_{25}Co_{75}$ alloy. The dynamics of this transient boundary determines the final magnetisation state as well as the speed of AO-HIS throughout the entire magnetic layer. The ability to understand the influence of nanoscale and transient inhomogeneities on ultrafast switching phenomena and more generally on phase transitions will open new routes for material design and excitation scenarios in future devices for transferring and storing information.

Deterministic switching of order parameters forms the foundation of modern data storage, memory, and computing technologies. Ultrashort laser pulses are ideal stimuli for driving these transitions, offering the potential for faster and more energy-efficient switching initiated at the quantum level[1–5]. In this context, manipulating magnetisation without any applied magnetic field is paramount. Besides spin-transfer torque, spin-orbit torque (SOT), and electric field-driven effects[6], the discoveries in the field of all-optical magnetisation switching (AOS)[7–9] have attracted a lot of attention. In particular, the all-optical helicity-independent magnetisation switching (AO-HIS) is a remarkable example where a single ultrashort laser pulse can completely reverse the magnetic order within a few picoseconds and on

nanometre length scales[10–12], without the need for an external magnetic field[13]. In gadolinium-based ferrimagnetic alloys such as GdFe or GdCo, AO-HIS is commonly attributed to the thermally-driven, distinct spin dynamics of the antiferromagnetically coupled transition metal (TM) and rare-earth (RE) sublattices[14]. Achieving AO-HIS requires a transfer of angular momentum between the two magnetic sublattices at a moment in time when one sublattice is close to full demagnetisation[15–17], which may even lead to the formation of a transient ferromagnetic-like state[18]. However, this widely accepted mechanism primarily accounts for the local and microscopic aspects of the process, overlooking any spatially inhomogeneous dynamics. While it has already been shown that femtosecond laser excitation can

---

[1]Max-Born-Institut für Nichtlineare Optik und Kurzzeitspektroskopie, Berlin, Germany. [2]Université de Lorraine, CNRS, Institut Jean Lamour, Nancy, France. [3]Leibniz-Institut für Kristallzüchtung, Berlin, Germany. [4]Technische Universität Berlin, Institut für Optik und Atomare Physik, Berlin, Germany. ✉e-mail: hennecke@mbi-berlin.de; schick@mbi-berlin.de

induce non-local dynamics along the lateral dimension, e.g., by an ultrafast spin-transfer between chemical inhomogeneities within an amorphous GdFeCo alloy[19], another significant inhomogeneity exists along the depth of a typical nanolayer system, in which the magnetic film is surrounded by distinct capping and seed layers. This inherent heterogeneity, although not originating from any structural or chemical inhomogeneities within the magnetic layer itself, gives rise to asymmetric excitation conditions which potentially affect the optically excited dynamics[20].

In this work, we experimentally follow the spatio-temporal evolution of AO-HIS along the depth of a typical magnetic heterostructure designed for optical switching, revealing that even within a sub-10 nm thin film of in-plane magnetised $Gd_{25}Co_{75}$ alloy, the switching is strongly inhomogeneous along the depth-axis. In particular, the speed and final result of the laser-induced switching process are governed by the transient formation and depth propagation of a domain-like region of reversed magnetisation. Our results suggest that the theoretically predicted criteria for successful AOS in magnetic thin-film systems need to be reconsidered with respect to the large spatial gradient of the magnetisation observed during the switching process.

## Results and discussion

### Time-resolved magnetic soft-X-ray spectroscopy

We investigate a typical nanometre-scale heterostructure consisting of an amorphous ferrimagnetic $Gd_{25}Co_{75}$ alloy with in-plane magnetic anisotropy, which exhibits deterministic picosecond AO-HIS induced by single laser pulses[21]. The thin $Gd_{25}Co_{75}$ layer (thickness 9.4 nm) is deposited on a Si substrate via magnetron sputtering, seeded at the bottom by a Ta layer (3.6 nm) and capped at the top with Cu (1.6 nm) and Pt (2.8 nm) to prevent oxidation of the ferrimagnetic film. For the studied $Gd_{25}Co_{75}$ alloy, the ferrimagnetic compensation point is above room temperature, where all measurements were carried out.

We enable an ultrafast and element-selective view on the dominant Gd sublattice magnetisation by femtosecond transverse magneto-optical Kerr effect (TMOKE) spectroscopy in the soft-X-ray spectral range. The experimental concept is illustrated in Fig. 1, combining a $\vartheta{-}2\vartheta$ reflectometry and spectroscopy setup (see panel a) in a pump–probe scheme. The Gd sublattice magnetisation is probed around the Gd $N_{5,4}$ resonance at ≈148 eV photon energy under an incidence angle of $\vartheta = 20°$, using ≤ 27 fs (full width at half maximum;

FWHM) short, broadband soft-X-ray pulses generated by a laboratory high-harmonic generation (HHG)-based light source[22,23]. The transient switching dynamics are excited by 27 fs (FWHM) short infrared (IR) pulses at 2.1 μm wavelength and probed by the magnetic asymmetry, i.e., the normalised difference of two spectra recorded for opposite directions of a saturating in-plane magnetic field applied to the sample (Fig. 1b).

Due to the distinct change of the complex refractive index along the Gd $N_{5,4}$ resonance in conjunction with interlayer reflection and interference effects, the soft-X-ray reflectivity and attenuation length vary strongly as a function of photon energy. The resulting variation of the probing depth along the resonance leads to a strong sensitivity of the asymmetry spectra to both the structural properties of the individual layers, i.e., their thickness, density, and roughness, as well as to the magnetisation distribution within the $Gd_{25}Co_{75}$ layer[20,24]. Fitting the transient asymmetry spectra with calibrated magnetic scattering simulations[25,26], therefore, allows determination of the transient magnetisation depth profiles (Fig. 1c). The broadband probing of the Gd $N_{5,4}$ resonance at a fixed angle of incidence enables capturing highly correlated spectral changes in a single, short acquisition. Moreover, it ensures constant excitation conditions, as the angle at which the IR pulses impinge on the sample remains unchanged during the entire experiment.

The recorded evolution of the magnetic asymmetry at the Gd $N_{5,4}$ resonance upon IR excitation is shown in Fig. 2, comparing the dynamics induced for different incident excitation fluences. The two depicted data sets for fluences of 5.0 and 6.0 mJ/cm² are recorded slightly below and above the threshold fluence required for a full reversal of the $Gd_{25}Co_{75}$ layer magnetisation.

For both excitation fluences, the time-resolved data show a non-uniform change of the magnetic asymmetry along the Gd $N_{5,4}$ resonance. Irrespective of whether the final state is switched or not, the asymmetry starts to gradually reverse from higher to lower photon energies on time scales ≥1 ps after excitation, see Fig. 2a. In the case of full switching (6.0 mJ/cm² excitation), the sign reversal of the entire spectrum is completed within a few picoseconds, whereas in the non-switching case (5.0 mJ/cm² excitation), the reversed part of the spectrum begins to relax back to the initial direction at ≈ 4.5 ps.

Integrating the magnetic asymmetry over different spectral regions around the main resonance peak (ROIs 1–3 in Fig. 2b) reveals

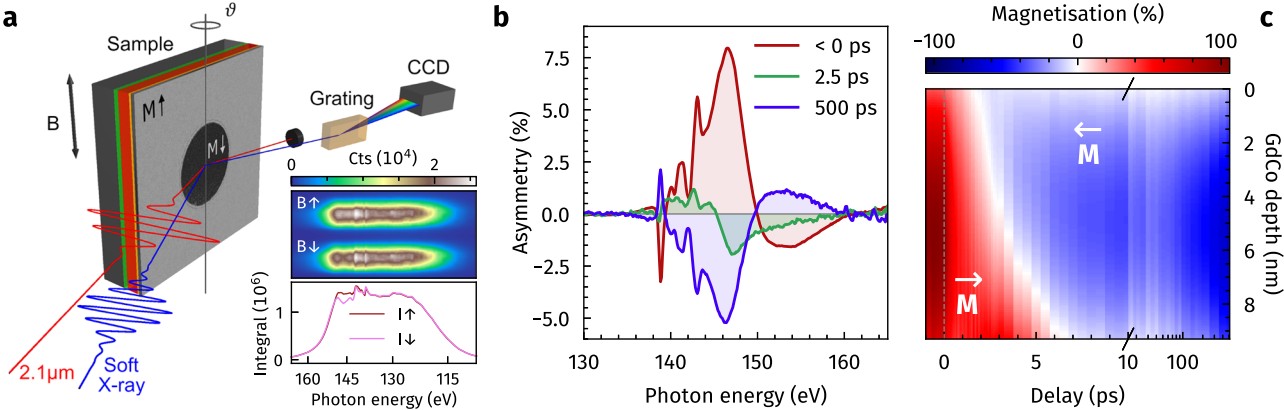

**Fig. 1 | Experimental concept. a** Schematic of the employed pump–probe technique. The all-optical helicity-independent magnetisation switching (AO-HIS) of an in-plane magnetised $Gd_{25}Co_{75}$ sample is driven by 27 fs (FWHM) short infrared (IR) laser pulses of 2.1 μm wavelength. The dynamics are probed at the Gd $N_{5,4}$ resonance by broadband transverse magneto-optical Kerr effect (TMOKE) spectroscopy, employing ≤ 27 fs (FWHM) short soft-X-ray pulses emitted by a high-harmonic generation (HHG) light source. Magnetic contrast is achieved by flipping a saturating in-plane magnetic field $B_{\uparrow,\downarrow}$ applied to the sample perpendicular to the $p$-polarisation axis of the probing soft X-rays, recording two spectra $I_{\uparrow,\downarrow}$ for

opposite magnetisation directions (see inset). **b** The magnetic asymmetry, corresponding to $(I_{\uparrow} - I_{\downarrow})/(I_{\uparrow} + I_{\downarrow})$, is recorded as a function of pump–probe delay. Due to wavelength-dependent probing depths in the vicinity of the atomic resonance, the spectra are sensitive to spatially inhomogeneous magnetisation dynamics along the depth of the $Gd_{25}Co_{75}$ layer, leading to peak shifts and non-uniform changes of the asymmetry. **c** Fitting the time-resolved spectra with magnetic scattering simulations enables the determination of the transient magnetisation distribution within the $Gd_{25}Co_{75}$ layer, linking ultrafast changes in the spectra to spatial changes of the depth profile.

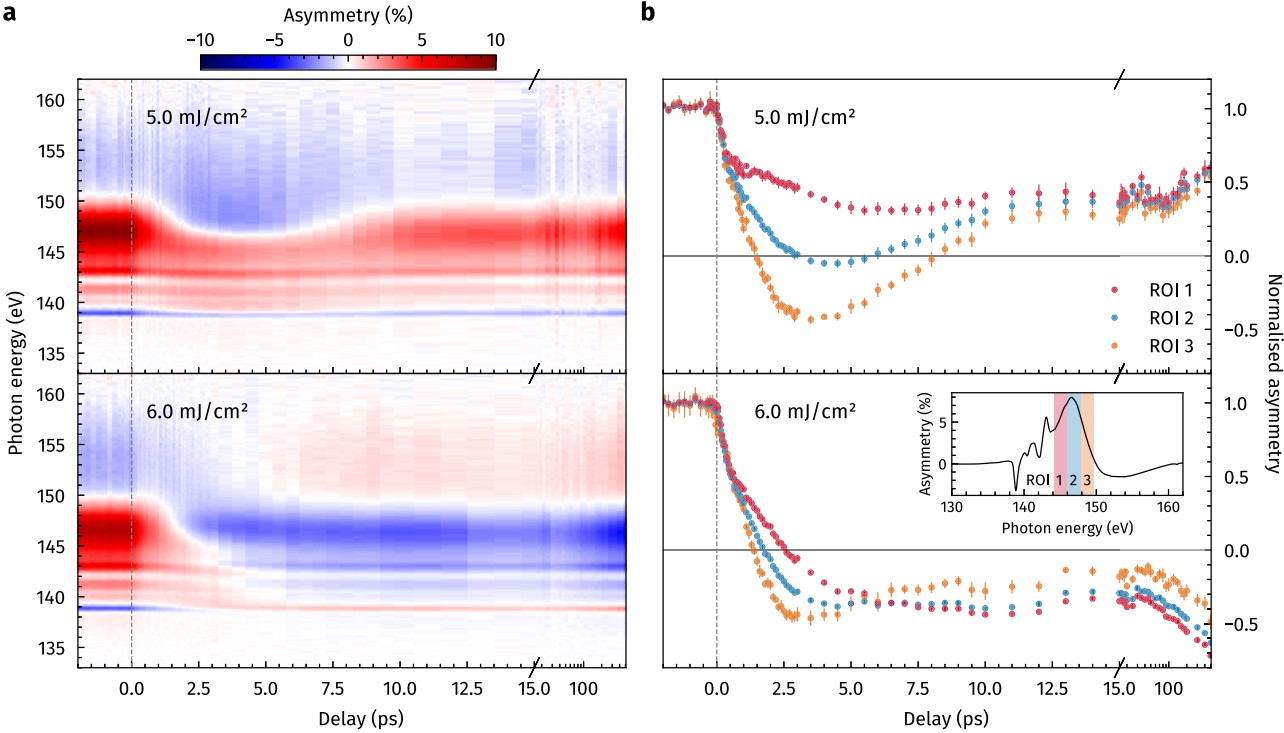

**Fig. 2 | Time-resolved magnetic soft-X-ray spectroscopy. a** Transient transverse magneto-optical Kerr effect (TMOKE) asymmetry (colour map) recorded at the Gd $N_{5,4}$ resonance under an incidence angle of $\vartheta = 20°$ as a function of pump–probe delay, comparing excitation below (5.0 mJ/cm²) and above (6.0 mJ/cm²) the threshold fluence for all-optical helicity-independent magnetisation switching (AO-HIS). For long delay times, the return to the initial unswitched state is obvious from the asymmetry spectrum for 5.0 mJ/cm² excitation, while switching to a state with reversed asymmetry occurs when the excitation fluence is raised to 6.0 mJ/cm².

**b** Normalised asymmetry time traces were obtained by integrating the time-resolved spectra over different regions of interest (ROIs) around the main resonance peak. The different colours correspond to the ROIs as indicated by the inset. The strong spectral dependence of the TMOKE observable causes transient differences of up to ≈ 80% normalised to the unpumped equilibrium state and even partial sign reversal of the spectrum (compare ROI 1 and 3 for 5.0 mJ/cm² excitation). The error bars are calculated from the standard error of the mean.

the high sensitivity of the TMOKE spectra to the probing soft-X-ray photon energy[27,28]. Due to the wavelength-dependent absorption length of the TMOKE observable, this is a clear indication of spatially inhomogeneous dynamics evolving along the depth of the magnetic layer[20], pointing at either partial or propagating AOS.

**Transient magnetisation depth profiles**

In order to quantitatively relate the observed spectral changes of the asymmetry to spatial changes in the magnetisation depth profile of the $Gd_{25}Co_{75}$ layer, the time-resolved TMOKE data is analysed via magnetic scattering simulations[25,26]. The results of this analysis are presented in Fig. 3 for selected delays before and after excitation.

Already before the pump pulse excites the sample (−1.0 ps), the fit converges for a non-uniform distribution of magnetisation within the $Gd_{25}Co_{75}$ layer, slightly decreasing towards the neighbouring Cu and Ta layers. This observation can be attributed to effects like interlayer diffusion, reducing the magnetisation at the interfaces with other non-magnetic materials[29,30]. Furthermore, static heating induced by the repetitive absorption of the pump radiation also affects the magnetisation of the unpumped state, which is particularly apparent by the fluence-dependent decrease towards the Cu interface.

For both excitation fluences and within the experimental uncertainty, the data shows that up to 0.5 ps after excitation, the $Gd_{25}Co_{75}$ layer demagnetises mostly homogeneously along the depth-axis, leading to a uniform change of the magnetisation profile relative to the unpumped state at −1.0 ps, where essentially no new magnetisation gradients are introduced (see dashed line in Fig. 3b). This is reasonable, as the simulated absorption profile of the 2.1 μm pump radiation

predicts only a slight optical excitation gradient within the $Gd_{25}Co_{75}$ layer itself (see inset of Supplementary Fig. 8b). At later times, however, the pump-induced change becomes strongly inhomogeneous, tilting the depth profile towards the top Cu interface. At this point in space, the magnetisation is completely quenched and subsequently reverses its sign.

The complete data set, evaluating the depth-resolved dynamics over the full range of pump–probe delays scanned in the experiment, is shown in Fig. 4. For a quantitative comparison of the dynamics occurring at the top, centre, and bottom of the $Gd_{25}Co_{75}$ layer, the magnetisation was spatially integrated over 1 nm-sized slices as indicated in Fig. 4a and normalised to the unpumped state before excitation, leading to the magnetisation time traces shown in Fig. 4b. On time scales > 0.5 ps, the depth-dependent dynamics lead to the formation of a highly inhomogeneous magnetisation profile, transiently dividing the 9.4 nm thin $Gd_{25}Co_{75}$ layer into domain-like regions magnetised in opposite directions at the top and the bottom of the thin film (red and blue colours in Fig. 4a). This bipolar state lasts for in total ≈ 4.5 ps (grey-shaded area in Fig. 4b), during which the region with opposite magnetisation direction starts to grow, corresponding to the propagation of the domain wall-like boundary between the two regions from the top towards the bottom of the magnetic layer. Depending on the excitation fluence, it either expands over the entire $Gd_{25}Co_{75}$ depth or relaxes back towards the initial unpumped direction (see Supplementary Movie 1 for an animation of the depth dynamics for the two fluences).

Intriguingly, the switching dynamics of the top 1 nm region appear to be identical for both excitation fluences within the first 3 ps after

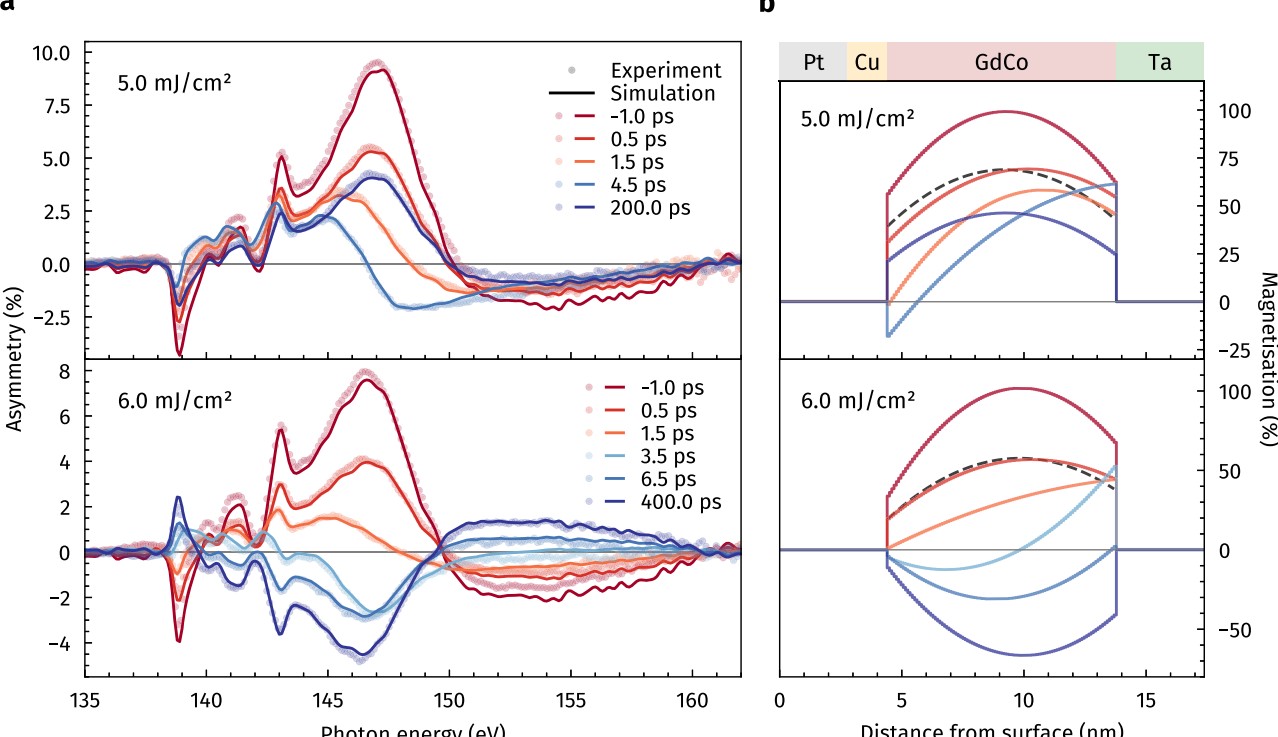

**Fig. 3 | Determination of the magnetisation depth profiles from the time-resolved transverse magneto-optical Kerr effect (TMOKE) data.**
**a** Experimentally recorded (dots) and fitted (lines) asymmetry spectra for selected pump–probe delays. The data was fitted by varying the magnetisation distribution within the $Gd_{25}Co_{75}$ layer. **b** Resulting magnetisation depth profiles for the respective excitation fluence and pump–probe delay. The black dashed line corresponds to the unpumped profile (−1.0 ps) scaled to the same average magnetisation value as the profile recorded at 0.5 ps.

excitation, irrespective of the final outcome of the entire film being switched or not switched. This observation indicates that already for 5.0 mJ/cm² excitation, which does not lead to a complete reversal of the $Gd_{25}Co_{75}$ layer, the switching threshold is nevertheless initially overcome within this confined surface-near region, resulting in early AOS dynamics that do not gain further amplitude and speed by increasing the excitation strength.

In contrast, the dynamics observed at the centre and bottom of the layer show a clear fluence-dependence on the same time scale (≤ 3 ps), with the demagnetisation scaling with the fluence, i.e., depending on the energy deposited into the magnetic layer. This suggests that after the initial ultrafast demagnetisation and partial reversal at the top, the thermal state as well as the remanent magnetisation in the not yet reversed part of the $Gd_{25}Co_{75}$ layer are crucial for whether the switched domain can further grow into the depth or is quenched again. Accordingly, the dynamics in the bottom slice shows a clear two-step behaviour in the case of 6.0 mJ/cm² excitation, allowing the determination of the time needed for the reversed domain to expand over the entire depth (9.4 nm) of the layer, which according to Fig. 4b is of the order of ≈ 4.5 ps, indicating an average velocity of the domain boundary of ≈ 2000 m/s.

## Tracing the origin of magnetisation switching

To further disentangle the fluence- and depth-dependence of the AO-HIS, the magnetisation depth profiling was carried out for a wide range of excitation fluences while keeping the pump–probe delay fixed at different times after excitation (Fig. 5a and Supplementary Fig. 4). The depth profiles recorded at late times (20 ps) exhibit simply a homogeneous transition from below to above the threshold fluence for AO-HIS ($F_{sw}$). In stark contrast, the data recorded at an earlier delay of 3 ps shows a strongly fluence-dependent formation of spatially

inhomogeneous switching dynamics, starting already at 4 mJ/cm², for which the surface-near region starts to transiently reverse. Again, a closer inspection of the magnetisation amplitudes at the early delay reveals that after the switching threshold is locally overcome at a certain depth within the top, surface-near region, the switching amplitude at this point in time gets saturated at ≈ −15% and does not further increase as a function of excitation fluence. Instead, increasing the fluence only leads to the initially switched region (≈ −15% amplitude at 3 ps) reaching further into the depth of the $Gd_{25}Co_{75}$ layer. By increasing the excitation fluence, more volume of the $Gd_{25}Co_{75}$ layer is driven into the direct laser-driven regime of AO-HIS, where the switching dynamics occur homogeneously within this region due to the local inter-sublattice angular momentum transfer as proposed by established theoretical models.

The initial formation of the switched region at the top of the layer can thus be understood by the material- and layer-dependence of the optical excitation within the heterostructure, with the largest amount of energy deposited in the Pt capping layer (see inset of Supplementary Fig. 8b). The strongly excited Pt layer acts as an effective source of heat and hot electrons, causing an additional indirect excitation reaching the $Gd_{25}Co_{75}$ layer from the top due to heat diffusion and ballistic electron transport across the highly conductive Cu layer[31]. Such hot electron pulses are known to induce ultrafast switching in GdFeCo-based heterostructures, even when the highly conductive spacer layer is thick enough to prevent direct optical excitation of the magnetic film[32,33]. Note that this observation is also in line with earlier depth-resolved studies on the ultrafast demagnetisation of a GdFe-based system, which have revealed that the strong absorption in Pt can even lead to an enhanced demagnetisation at the bottom of the magnetic film, when the Pt is used as a seed instead of a capping layer[20]. It further demonstrates how strongly the dynamics of the

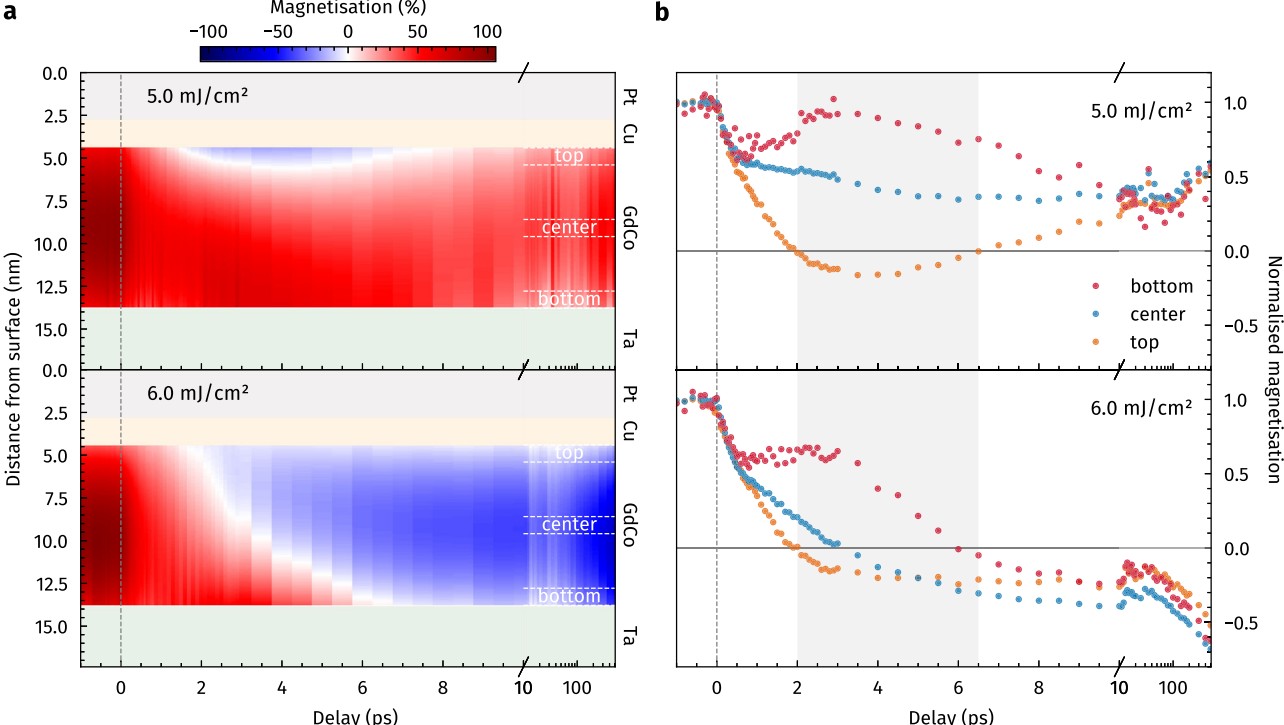

**Fig. 4 | Depth-resolved magnetisation dynamics. a** Transient magnetisation distribution (colour map) within the $Gd_{25}Co_{75}$ layer as a function of pump–probe delay. The white area corresponds to zero magnetisation, i.e., a transient boundary between the two regions with opposite magnetisation directions. **b** Spatially integrated and normalised magnetisation time traces, comparing the dynamics within 1 nm thin regions at the top, centre, and bottom of the $Gd_{25}Co_{75}$ layer as indicated in (**a**). The grey-shaded area corresponds to the time interval during which the layer is transiently split into oppositely magnetised domain-like regions at the top and the bottom of the magnetic film.

magnetic layer can be influenced by the heterostructure design, in particular, the choice of the surrounding layers, which also opens a potential route for tuning, e.g., directionality and energy efficiency of the switching.

In theory, the excitation fluence could be increased until the entire depth of the layer is directly driven into the AO-HIS regime. In reality, the slight gradient in optical excitation of the $Gd_{25}Co_{75}$ layer itself, in combination with the highly absorbing Pt layer, leads to an excessive heat load at the surface of the magnetic layer, degrading the sample when it is exposed to fluences above 6.0 mJ/cm² for a prolonged period of time. Hence, the full, persistent switching of the entire film, observed above the clearly visible threshold fluence of 5.5 mJ/cm² ($F_{sw}$), can only be achieved via a second, non-local process, which drives the boundary between the two magnetic regions towards the bottom of the magnetic layer.

To better understand the nature of this non-local switching mechanism, we will now discuss the role of different phenomena that can potentially lead to the propagation of the AO-HIS switching boundary into the depth of the magnetic layer.

Diffusive two-temperature model (2TM) simulations[25] predict the electronic and phononic heat baths to be almost fully equilibrated already after $\approx 2$ ps (see Supplementary Fig. 8). Furthermore, at the time when the bottom of the $Gd_{25}Co_{75}$ layer starts to reverse its magnetisation (>3 ps in Fig. 4b), the temperatures are already decreasing within this region due to efficient heat dissipation into the Si substrate. Hence, the simulations suggest that picosecond heat transport along the depth of the heterostructure is not able to drive the lower regions of the $Gd_{25}Co_{75}$ layer into the AO-HIS regime via thermal demagnetisation and, therefore, cannot explain the delayed onset of the switching observed within this region.

Instead, the continuous heat dissipation into the Si substrate leads to the formation of a long-lived thermal gradient of 10–30 K between the hotter surface-near region and the colder bottom of the $Gd_{25}Co_{75}$ layer (determined at the boundaries of the grey-shaded area in Supplementary Fig. 8), persisting throughout the time interval in which the switched region expands. Such temperature gradients are known to cause a thermally-driven domain wall motion, either due to spin or magnon currents generated by the spin Seebeck effect[34,35] or to maximise the domain wall entropy[36,37], playing a crucial role in multiple-pulse induced AOS observed in ferromagnetic thin films and multilayers[38,39]. In case of ferrimagnets, the direction of the domain wall motion has been predicted to depend on whether the temperature of the material is below or above the angular momentum compensation point ($T_{comp}$) of the two magnetic sublattices, driving the domain wall either towards the colder or hotter regions, respectively[40]. Although this mechanism could possibly explain the fluence-dependent propagation of the observed boundary, where direction and velocity depend on the depth-dependent temperatures transiently reached after electron-phonon thermalisation, its role in view of the highly non-equilibrium energy and heat distribution on femto- to picosecond time scales needs to be further investigated.

Spatially inhomogeneous AO-HIS dynamics have also been theoretically predicted in synthetic ferrimagnets, namely, Co/Gd bilayers[41]. Here, the proposed mechanism is based on exchange scattering across neighbouring atomic layers. This would lead to the nucleation of a front of reversed Co magnetisation at a Co/Gd interface, which can subsequently propagate away from the interface through the Co layer in a few picoseconds and over several nanometres. The exchange scattering with nearest neighbours enables AO-HIS far away from the compensation point of the total bilayer at a comparable speed

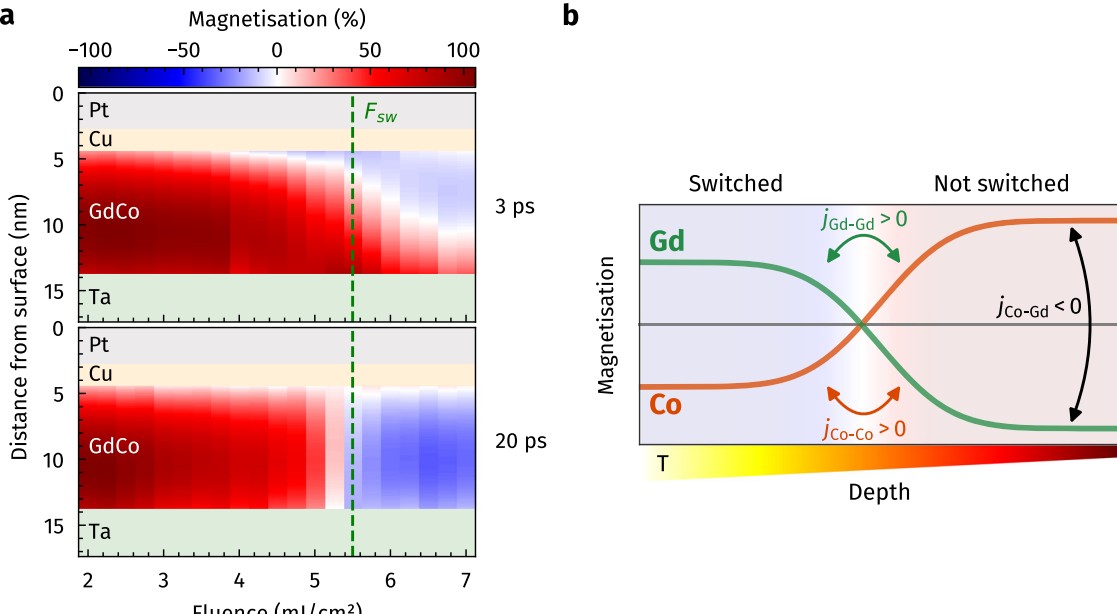

**Fig. 5 | Fluence-dependence and phenomenological picture of the propagating all-optical helicity-independent magnetisation switching (AO-HIS). a** Fluence-dependence of the transient magnetisation distribution (colour map) within the $Gd_{25}Co_{75}$ layer, comparing different times after excitation. The magnetisation depth profiles are obtained by fitting transient transverse magneto-optical Kerr effect (TMOKE) spectroscopy data recorded at fixed pump–probe delays (3 and 20 ps) as a function of excitation fluence (2.0 to 7.0 mJ/cm²). The dashed green line indicates the threshold fluence $F_{sw}$ = 5.5 mJ/cm², which is required for a full reversal

of the $Gd_{25}Co_{75}$ layer magnetisation. **b** Schematic of the growth of a domain-like directly-switched region (left, blue) into the depth of the sample (right, red). The sublattice magnetisation of Co and Gd atoms is sketched by the green and orange curves. Exchange between Co and Gd, as well as between Gd–Gd and Co–Co at the propagating boundary between switched and unswitched regions, is indicated by arrows. The coloured wedge symbolises the quasi-static temperature gradient from hot (white/yellow) to cold (red/black) as calculated from the diffusive two-temperature model (2TM).

($\approx$1000 m/s, obtained from Fig. 3 of ref. 41 assuming a Co monolayer thickness of 0.2 nm) as observed in our experiment ($\approx$2000 m/s). The proposed mechanism, however, does involve an initial complete demagnetisation of the entire Co layer induced by the direct optical excitation. In contrast, our experimental results reveal a sizeable magnetisation of the Gd sublattice in the lower, not directly-switched region of the $Gd_{25}Co_{75}$ layer, see Fig. 5b, requiring the switching mechanism to overcome a large magnitude of remanent magnetisation.

The propagation speed of the observed boundary ($\approx$2000 m/s) thereby lies within the top range of the fastest velocities of domain wall motion that have so far been realised experimentally in ferrimagnetic GdCo and GdFeCo wires, driven by current-induced SOT when the RE and TM sublattices are close to angular momentum compensation ($\approx$1300–2000 m/s)[42,43]. This suggests that the reduced net magnetisation and increased domain wall velocity in the vicinity of the ferrimagnetic compensation point might play a decisive role in the second, non-local switching mechanism, facilitating an efficient growth of the switched region over the entire depth of the layer.

Our results show that even in sub-10 nm thin alloy films embedded into a typical nanolayered heterostructure, the combination of local and non-local processes is essential for achieving complete AO-HIS. We find that asymmetric excitation conditions can drive parts of the ferrimagnetic layer into direct switching, but in other regions, the laser-driven demagnetisation is not sufficient to enter the narrow transient regime for AO-HIS. Instead, a switching front travelling into the depth of the material is crucial to achieve complete AO-HIS under suitable conditions. While thermal gradients are present, we rule out simple heat transport into the depth of the film as the essence of the switching mechanism, based on the observed time scale of the switching front propagation. Nevertheless, the thermal gradients and additional channels for exchange at the boundary between switched and

unswitched regions, see Fig. 5b, provide a key ingredient for alternative, propagation-based microscopic mechanisms. While they are present on the relevant picosecond time and nanometre length scales, their actual influence on the switching process needs to be clarified in future studies, including a theoretical treatment. To this end, the variation of the stacking sequence, the capping and seed layer materials, as well as the composition and thickness of the magnetic film, will enable direct control over the layer-dependent absorption and, accordingly, the formation, size, and direction of excitation gradients and related transport phenomena. Hence, a systematic investigation of AO-HIS for well-considered heterostructure designs will shed light on the conditions under which non-local, propagation-based switching can be observed, as well as on the actual mechanism driving the propagation of the transient boundary. Disentangling longitudinal and transverse spin dynamics might provide further insights into the nature and shape of the observed boundary, as well as the microscopic processes occurring on the different time scales of direct and propagation-based switching. Furthermore, modelling the observed inhomogeneous dynamics, e.g., employing atomistic spin dynamics simulations, could help to identify the key ingredients for the occurrence of either local or non-local switching processes and also to transfer the obtained insights to switching phenomena in other types of magnetic materials.

In summary, our findings add a new dimension to the parameter space for achieving AO-HIS as an ultrafast and efficient route to deterministic magnetisation switching without external magnetic fields. In particular, the fluence-dependence of the speed of AO-HIS averaged over the full thickness of a ferrimagnetic layer can be well explained by our findings as a transition between local, direct switching and the non-local effect of a propagating switching front. Accordingly, tailoring the excitation conditions across the magnetic heterostructure can significantly improve the overall speed and energy

efficiency of AO-HIS. In perspective, impulsively triggered switching processes such as AO-HIS must generally be considered as inhomogeneous on ultrafast time scales and over distances as short as a few nanometres along the depth of typical thin-film samples. This becomes particularly relevant when the active layer is embedded in a heterostructure, as the surrounding layers can easily cause an inherent inhomogeneity along the depth, leading to asymmetric excitation conditions that facilitate the formation of inhomogeneous dynamics or even determine their direction. We expect these findings to be relevant for phase transitions driven via non-equilibrium conditions in general. In this context, we would like to conclude by noting that our method for obtaining depth resolution is merely based on a spectroscopic observable and thus not limited to the investigation of magnetic effects. Therefore, we anticipate its application also for a wider range of non-magnetic phenomena. The ability to resolve such dynamics in time and space, in particular with element-specificity, can resolve puzzling observations of distinctively different dynamics for the same material systems, but, e.g., for different sample geometries.

## Methods

### Time- and angle-resolved magnetic soft-X-ray spectroscopy

The static and time-resolved measurements are carried out in a combined $\vartheta$–$2\vartheta$ reflectometry and spectroscopy setup, enabling both transmission and specular reflection geometries for $\vartheta$ angles between 0 and 45° with respect to the sample plane.

The probing soft X-rays are provided by a laboratory light source based on HHG, which is operated with the noble gas argon ($\approx 280$ mbar gas pressure). The HHG process is driven by a 2.1 μm, 27 fs (FWHM) high average power infrared (IR) optical parametric chirped pulse amplification (OPCPA) system and generates $p$-polarised $\leq 27$ fs soft X-ray pulses at 10 kHz repetition rate, covering a broad and continuous spectrum in the range of 100–200 eV (source photon flux $\approx 10^9$ photons/eV/s @ 150 eV)[23]. The soft X-ray pulses are focused into the rotation centre of the reflectometer upon reflection by a toroidal mirror placed under a grazing angle of 3° at its incident/exit distance of 1000 mm, leading to a focal spot size of $85 \times 80$ μm$^2$ (FWHM). As the horizontal projection of the probe spot on the sample plane scales with $1/\sin\vartheta$, the resulting spatial footprint is $250 \times 80$ μm$^2$ (FWHM) for a reflection angle of $\vartheta = 20°$, at which the time-resolved experiments are carried out. The soft X-rays reflected by the sample are guided over a variable line spacing (VLS) grating with a central line density of 1200 l/mm (Hitachi 001-0437), which is placed under a grazing angle of 3° at its focal distance of 237 mm. The first and higher order reflection of the VLS grating horizontally disperses and focuses the soft X-ray spectrum on an in-vacuum CCD camera (Greateyes GE 2048 512 BI), resulting in a photon energy resolution of $\approx 0.3$ eV. The CCD is protected from parasitic stray light by a 200 nm thin silver filter, which is opaque for the 2.1 μm pump beam, but transmissive for radiation in the range of 100–200 eV. After background subtraction, the CCD images are vertically integrated, and the horizontal photon energy axis of the first order reflection is derived from the reflection grating formula, assigning a single wavelength $\lambda$ to each pixel of the CCD[20]:

$$\lambda = \frac{1}{g} \left[ \sin\left( \arctan \frac{1}{\frac{1}{\tan\vartheta_{\text{in}}} + \frac{\Delta x}{r}} \right) + \sin\vartheta_{\text{in}} \right], \qquad (1)$$

with $g$ corresponding to the central line density of the grating, $\vartheta_{\text{in}}$ to the grating angle of incidence, $r$ to the distance between camera and grating, and $\Delta x$ to the position of each pixel ($13.5 \times 13.5$ μm$^2$) on the CCD relative to the specular (zeroth order) reflection. To increase the accuracy of the photon energy axis, the uncertainties in the determination of the exact CCD distance and grating angle are corrected by calibrating the spectrometer to a reference measurement at the Gd $N_{5,4}$ resonance and by taking the second-order reflection of the grating into account[20].

For the pump–probe measurements, $\approx 10\%$ of the $p$-polarised 2.1 μm IR beam is coupled out before the HHG cell and guided over an optical delay line, serving as an intrinsically synchronised pump beam for exciting the sample. The 27 fs (FWHM) pump pulses are focused onto the sample using a refractive optical lens and hit the sample almost collinearly with the soft X-ray pulses, with the angle of incidence being 3.3° more grazing compared to the probe. By tuning the lens position, the pump spot size is adjusted to be at least four times larger in both dimensions compared to the probe spot size, clearly overfilling the probed area, which thus can be considered as homogeneously pumped. The spatial overlap between pump and probe pulses is initially achieved by guiding both pulses through a pinhole placed at the soft X-ray focus, and fine-tuned directly on the sample by scanning the pump spot position with a motorised overlap mirror, optimising for the largest pump-induced change. The overall temporal resolution of the pump–probe measurements is limited by the pulse duration of the pump and probe pulses as well as the enclosed angle between the two beams, and corresponds to $\leq 40$ fs.

The magnetic contrast is achieved via transverse magneto-optical Kerr effect (TMOKE) spectroscopy, recording two spectra $I_{\uparrow,\downarrow}$ for opposite and saturating in-plane magnetic fields of $B_{\uparrow,\downarrow} = \pm 40$ mT applied to the sample, which alternates between both directions perpendicular to the $p$-polarisation axis of the probing soft X-ray pulses and restores the initial magnetisation state after each pump–probe cycle. The evaluated observable is the magnetic asymmetry, i.e., the normalised difference between the two spectra as a function of pump–probe delay (see Fig. 2). Employing such a difference scheme to obtain magnetic contrast is reasonable, as the studied $Gd_{25}Co_{75}$ sample possesses an in-plane magnetic anisotropy with an easy-axis parallel to the surface, along which the magnetisation has no preferred direction[21]. Thus, the magnetic properties of the sample and the induced dynamics are entirely independent of whether the magnetisation is initially aligned parallel or antiparallel to a particular direction along this easy axis.

To minimise the influence of intensity fluctuations of the HHG source on the magnetic contrast, each of the recorded spectra $I_{\uparrow,\downarrow}$ is normalised to a spectral range slightly below the atomic resonance, where the TMOKE contribution to the reflectivity signal is zero. The magnetic asymmetry $A$ is then calculated as $A = (I_{\uparrow} - I_{\downarrow})/(I_{\uparrow} + I_{\downarrow})$.

The static angle-resolved TMOKE measurements used for the initial calibration of the magnetic scattering simulations as well as the determination of the equilibrium magnetisation profile are carried out for angles of incidence ranging from $\vartheta = 2.5°$ up to 42.5° in 2.5° steps. Due to the exponentially decaying reflectivity as a function of $\vartheta$, the illumination times are adjusted accordingly in order to achieve a reasonable signal-to-noise ratio over the full $\vartheta$ range.

All time-resolved measurements are carried out at $\vartheta = 20°$ with an acquisition time of 30 s per magnetic field direction, recording an image pair ($I_{\uparrow,\downarrow}$) for each pump–probe delay. The delay scans are recorded for excitation fluences of 5.0 and 6.0 mJ/cm$^2$, which is slightly below and above the switching threshold fluence. For the system studied, the threshold value of $\approx 5.5$ mJ/cm$^2$ is determined according to the magnetisation state observed at 500 ps after excitation (see Supplementary Fig. 4). As the dynamics observed at such late times are well within the relaxation regime, they can be considered to reflect the final outcome of the $Gd_{25}Co_{75}$ layer magnetisation. This is either switched or non-switched, i.e., relaxed in the opposite or initial magnetisation direction, respectively. At much longer delays, the external magnetic field will reset and saturate the magnetisation of the sample in a defined direction independent of the laser fluence. The transient TMOKE asymmetry shown in Fig. 2 is averaged over eight delay scans in case of 5.0 mJ/cm$^2$, and ten delay scans in case of 6.0 mJ/cm$^2$ excitation, leading to a total measurement time of 15 and 19 hours, respectively.

## Magnetic scattering simulations

The simulations used to fit the experimental TMOKE data are carried out using the udkm1Dsim toolbox[25,26], calculating the polarisation- and magnetisation-dependent soft-X-ray reflectivity from the element-specific atomic and magnetic form factors of the individual layers in the heterostructure, taking into account any interlayer reflection and interference effects, as well as the depth-dependent variation of both the structural and magnetic properties. The structural parameters of the sample are initially determined via hard-X-ray specular reflecto-metry (XSR) (see corresponding Methods section). The spectral dependence of both real and imaginary parts of the form factors of the $Gd_{25}Co_{75}$ alloy at the Gd $N_{5,4}$ resonance (see Supplementary Fig. 1) is taken from reference measurements on stoichiometrically similar GdFe and GdCo membrane samples, which are based on an inter-ferometric method that allows the simultaneous retrieval of amplitude and phase contrast from the interference of circularly polarised soft X-rays[44]. The form factors of Pt, Cu, Ta, and Si in the respective spectral range, which is located far from their atomic resonances, are obtained from tabulated values[45].

Static angle-resolved TMOKE spectroscopy is employed to determine the equilibrium magnetisation profile of the $Gd_{25}Co_{75}$ layer and to calibrate the amplitude of the $Gd_{25}Co_{75}$ atomic form factors, which are taken from the reference measurements, for the studied sample system. The latter is required, since different sample growth parameters can change the properties of the amorphous alloy (e.g., density, absorption), affecting the overall amplitude of the form factor spectrum without changing the relative spectral dependence along the resonance. To this end, the angle-resolved asymmetry spectra are fit-ted with the simulations, keeping the structural layer parameters fixed and varying the magnitude and distribution of magnetic moment within the $Gd_{25}Co_{75}$ layer as well as a constant scaling factor applied to the form factor spectrum. The magnetisation profile is thereby allowed to be inhomogeneous, modelled by a second-order polynomial dis-tribution along the sample normal. In agreement to earlier work[20], this is the simplest analytical model to allow a consistent description of both static and time-resolved data, reflecting the asymmetric layer structure of the sample with distinct cap and seed layers. To limit the computational complexity, the angle-resolved fit of the asymmetry data was carried out only for a subset of seven $\vartheta$ angles (10, 15, 20, 25, 30, 35, 40°). The resulting good agreement between experimental data and simulations, as well as the obtained equilibrium magnetisation profile, are shown in Supplementary Fig. 3. The residuals between measured and simulated asymmetry, which are more prominent for larger $\vartheta$ angles, can be attributed both to the drastically decreasing reflectivity and therefore worse statistics, as well as to the decreasing footprint of the soft X-rays on the sample surface, increasing the sensitivity to lateral inhomogeneities. As the main goal of the study is to analyse the dynamics recorded at $\vartheta = 20°$, the static asymmetry fits are carried out with a higher weight on this particular angle, most accurately describing the probing volume of the time-resolved measurements.

The time-resolved asymmetry spectra are fitted for each pump–probe delay, varying the transient magnetisation depth profile of the $Gd_{25}Co_{75}$ layer as the only free parameter, while keeping the structural parameters and form factors fixed as determined from the static XSR and TMOKE measurements. This procedure is justified, as the TMOKE observable is insensitive to any potential structural or purely electronic dynamics, which is confirmed by scattering simula-tions based on the experimental, non-magnetic reflectivity spectra (see Supplementary Fig. 7). An assessment of the fitting routine's sig-nificance confirms that the experimental data, especially in the time interval where the magnetisation of the $Gd_{25}Co_{75}$ layer is partially reversed, cannot be accurately described by assuming spatially homogeneous spin dynamics (see Supplementary Fig. 5). Furthermore, offsetting the fitted magnetisation profiles by > 2% is inconsistent with the experimental data, illustrating the sensitivity of the analysis (see Supplementary Fig. 6).

## Hard-X-ray specular reflectometry

The structural, non-magnetic parameters of the sample, i.e., the thick-ness, density, and roughness of the individual layers, which are required for an accurate simulation of the TMOKE asymmetry, are independently and accurately determined via XSR, using a commercial high-resolution X-ray diffractometer (Rigaku SmartLab 9 kW). Behind a precollimating parabolic multilayer mirror, an asymmetric two-fold Ge(220) channel-cut crystal selects the Cu K$\alpha_1$ line (8.047 keV) and simultaneously col-limates the primary beam in the sample reflection plane to about $\Delta\vartheta = 0.008°$. Subsequently, the useful beam is narrowed to 0.1 mm. The specularly reflected X-ray beam from the sample passes through a 0.1 mm exit slit and is subsequently detected by a single photon counting area detector (HyPix-3000) using a virtual slit of 0.3 mm. The experi-mental setup enables a very good angular resolution and a very low scattering background, so that a large dynamic intensity range is achieved. The experimentally determined X-ray reflectivity is fitted within the framework of dynamical scattering theory, applying Parratt's recursive formalism[46] implemented in the software package RCRefSimW[47]. The experimental data, the corresponding fit and the obtained structural parameters are shown in Supplementary Fig. 2. Note that the densities of the amorphous layers can vary slightly along the depth due to effects like interlayer diffusion of atoms, growth inho-mogeneities or oxidation processes, and may therefore deviate from the nominal values for the respective elements. To approximate such effects, the Pt, Cu, and Ta layers are each divided into two sublayers, whose thicknesses, densities, and roughnesses are allowed to be dif-ferent, leading to a better fit of the XSR and angle-resolved TMOKE data.

## Two-temperature model simulations

The diffusive 2TM simulations are performed using the udkm1Dsim toolbox[25], solving a set of two coupled differential equations, which describe the depth-resolved transient evolution of electron ($T_e(z,t)$) and phonon ($T_p(z,t)$) temperatures, including one-dimensional heat diffusion along the sample normal ($z$):

$$c_e\rho\frac{\partial T_e}{\partial t} = \frac{\partial}{\partial z}\left(k_e\frac{\partial T_e}{\partial z}\right) - G_{ep}\left(T_e - T_p\right) + S$$
$$c_p\rho\frac{\partial T_p}{\partial t} = \frac{\partial}{\partial z}\left(k_p\frac{\partial T_p}{\partial z}\right) + G_{ep}(T_e - T_p) \tag{2}$$

Here, $c_e(z,T_e)$ and $c_p(z)$ denote the material-dependent electron and phonon specific heat capacities of each layer, $k_e(z,T_e,T_p)$ and $k_p(z)$ the respective thermal conductivities, and $G_{ep}(z)$ the electron-phonon coupling constants. $c_p(z)$ and $k_p(z)$ are thereby assumed to be independent from $T_p$, while $c_e(z,T_e) = \gamma(z)T_e(z,t)$ and $k_e(z,T_e,T_p) = k_e^0(z)\frac{T_e(z,t)}{T_p(z,t)}$ with the material-specific electronic heat capacity coefficient $\gamma(z)$ and equilibrium heat conductivity $k_e^0(z)$[31]. The optical excitation is taken into account by $S(z,t)$, describing the pump laser energy absorbed by the electron heat bath. To this end, the depth-dependent differential absorption profile of the 2.1 μm pump pulse is first calculated using a multilayer formalism within the udkm1Dsim framework, taking interlayer reflection and interference effects into account, resulting from the material-dependent complex refractive index change at each interface of the heterostructure (see the inset of Supplementary Fig. 8b). To account for the pulsed excitation in the time-resolved 2TM simulations, the calculated absorption profile is enveloped by a Gaussian distribution in time domain, with 27 fs (FWHM) corresponding to the experimental pump pulse duration. The material-specific parameters used for the 2TM (see Supplementary Table 1) are obtained from literature values for the thermal and optical properties of the individual layers in the heterostructure.

## Data availability

All data needed to evaluate the conclusions of this study are presented in the main article and/or the Methods section, and have been deposited in a public Zenodo repository (https://doi.org/10.5281/zenodo.16800379)[48].

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

## Acknowledgements

We gratefully acknowledge funding by the German Research Foundation (DFG, Germany) through CRC/TRR 227 project A02 (project ID 328545488, C.v.K.S., S.E.), by the Leibniz Association through the Leibniz Junior Research Group J134/2022 (M.M., D.S.) and by the European Union through ERDF projects 1.6/03 (M. Hennecke, S.E.) and 1.8/15 (M. Schmidbauer). This work has been supported by the ANR through the SLAM project (ANR-23-CE30-0047, G.M., S.M., M. Hehn) and the France 2030 government grants PEPR Electronic EMCOM (ANR-22-PEEL-0009, J.-X.L., S.M.) and PEPR SPIN (ANR-22-EXSP-0002, G.M., S.M., M. Hehn), as well as the MAT-PULSE-Lorraine Université d'Excellence project (ANR-15-IDEX-04-LUE, J.-X.L., G.M., S.M., M. Hehn). The experiments were carried out at the NanoMovie Application Laboratory at the Max Born Institute, which was established with funding from the European Union through the ERDF project 1.8/10.

## Author contributions

M. Hennecke, D.S., S.M., M. Hehn and S.E. originally conceived the experimental idea. M. Hennecke carried out the static and time-resolved TMOKE measurements, the data treatment, the simulations and the magnetisation depth profiling. J.X.L., Z.G. and M. Hehn grew and characterised the sample. M. Schmidbauer carried out the hard-X-ray reflectometry measurements and the determination of the structural parameters of the sample. T.P.H.S., L.E., and M. Schnuerer contributed to the development of the experimental setup. M. Hennecke, D.S., S.M., M. Hehn, G.M. J.-X.L., Z.G., M.M., C.v.K.S. and S.E. contributed to the discussion and interpretation of the results. M. Hennecke and D.S. prepared the manuscript with contributions from C.v.K.S., S.M., M. Schmidbauer and S.E. All authors commented on the manuscript.

## Funding

## Competing interests

The authors declare no competing interests.
