## [Transparent Peer Review file · Nature Communications]

Transient domain boundary drives ultrafast magnetisation reversal

Corresponding Author: Dr Martin Hennecke

Version 0:

Reviewer comments:

Reviewer #1

(Remarks to the Author)

The authors have investigated the mechanism of all-optical helicity-independent magnetization switching (AO-HIS) in GdCo multilayer structure using pump-probe technique. The space and temporal evolutions of magnetization have been measured through MOKE. They show that the AO-HIS is governed by the propagation of the domain-like region in the depth direction. It is commendable that non-uniform magnetization reversal was clearly observed. However, non-uniform reversal itself has already been reported (as ref. 37 in the manuscript) and thus, it cannot be said to be a novel phenomenon. In order to publish this manuscript in Nature Communications, I think that the following points need to be clarified.

1. Whether both up-to-down and down-to-up magnetization reversals proceed with the same mechanism is important. But in this manuscript, I can see only one of them.
2. The Pt cap layer plays an important role. What is the reversal mechanism when Pt is used as the bottom instead of the cap? In addition, how is the switching behavior affected when the cap layer is changed to a different material, especially one with low absorption?

The followings are my questions and comments which will help to revise the manuscript.

1. Which sublattice magnetization is dominant at room temperature, Co or Gd? Why did you probe the Gd sublattice magnetization?

2. In Fig. 3(b), magnetization depth profiles are shown. What is the origin of the CoGd magnetization decrease near the Cu/CoGd and CoGd/Ta interfaces for unpumped state (-1.0 ps)?

Reviewer #2

(Remarks to the Author)

The study by Hennecke et al. provides a transformative perspective on the mechanisms underlying all-optical helicity-independent switching (AO-HIS) in ferrimagnetic GdCo alloys, revealing that magnetization reversal is driven by a transient domain boundary propagating along the depth of the magnetic layer. By combining time-resolved soft-X-ray spectroscopy (TMOKE) with magnetic scattering simulations, the authors demonstrate that the process is spatially inhomogeneous, challenging the prevailing view of AO-HIS as a purely local, thermally driven process. The depth-resolved dynamics show a bi-domain magnetization state emerging post-excitation, with a boundary moving at ~2000 m/s, a speed comparable to current-induced domain wall velocities in ferrimagnets. This propagation is not explained by thermal diffusion alone, as supported by two-temperature model (2TM) simulations, and instead implicates non-local mechanisms such as spin-Seebeck effects or exchange interactions.

The experimental design is rigorous, with structural parameters validated via hard X-ray reflectometry and sensitivity analyses confirming the necessity of inhomogeneous models over homogeneous switching assumptions. Fluence-dependent measurements reveal a threshold behavior (5.5 mJ/cm²), below which partial reversal relaxes and above which the boundary propagates to completion. These findings are bolstered by spectral asymmetry data and late-time

magnetization profiles, ensuring robust validation of the proposed mechanism. The study's strengths lie in its innovative methodology, which provides the first direct observation of depth-resolved AO-HIS dynamics, and its quantitative analysis linking domain wall velocities to established spintronic phenomena.

While the work is focused on GdCo alloys, the implications extend to broader research on ultrafast phase transitions and nanoscale magnetism. A minor limitation is the unresolved theoretical specifics of the boundary propagation mechanism, which the authors acknowledge and suggest for future exploration. The presentation is clear, with well-constructed figures and supplementary data (e.g., Extended Data) offering comprehensive validation. Although the textual description suffices to convey key insights, the supplementary video enhances understanding of domain dynamics.

The significance of the study lies in updating the understanding of AO-HIS, emphasizing spatially inhomogeneous dynamics critical for ultrafast switching. This work not only advances fundamental knowledge but informs the design of energy-efficient spintronic devices. Recommended revisions include briefly addressing the potential generality of the mechanism to other materials and clarifying theoretical open questions. Overall, the manuscript meets the criteria of Nature Communications for technical rigor, novelty, and broad relevance, warranting publication after minor revisions to enhance contextual discussion and mechanistic clarity.

Reviewer #3

(Remarks to the Author)

This study explores ultrafast all-optical helicity-independent magnetisation switching (AO-HIS) in sub-10 nanometre GdCo alloy films. Using femtosecond X-Ray transverse magneto-optical Kerr (TMOKE) spectroscopy at the Gd resonance, the authors track the absorption depth-resolved evolution of magnetisation. The results reveal that switching is not uniform across the film; rather, it initiates near the surface and propagates inward via a transient domain boundary. This non-local switching process is driven by the platinum capping layer, which absorbs the infrared pump energy and transfers heat and hot electrons into the underlying GdCo, triggering magnetisation reversal. These findings highlight the critical role of interface-driven dynamics and challenge the conventional view of AO-HIS as a purely local phenomenon.

Although questions regarding non-local dynamics in GdFeCo have been raised previously—most notably by Graves et al. (Nature Materials, 2013), who showed they can originate from the sub-10 nm patches of REs and TMs—this work provides a compelling and technically advanced demonstration using state-of-the-art soft X-ray spectroscopy, supported by detailed modelling.

I have a few questions for the authors:

- The fitting procedure appears to involve a large number of parameters. How unique and robust are the spatial magnetisation profiles derived from the simulations?
- What would happen if a different capping material were used instead of platinum? How critical is Pt for the observed dynamics? What if gold is used instead?
- To what extent is the observed depth profile governed by interfacial effects versus the overall thickness of the GdCo layer? Could this be systematically studied, either experimentally or through simulations?
- Do the results support or suggest the possibility that hot electron currents can be used to induce magnetisation switching in GdCo?
- The study reveals spatial inhomogeneity along the film depth. In Graves et al., lateral inhomogeneity with characteristic lengths of ~10 nm was also observed. Can the authors comment on the significance of this comparison? Is there any link between lateral and depth inhomogeneities in these systems?
- Does it happen that the suggested inhomogeneous mechanism dominates the thermally-induced uniform switching in GdFeCo? When can it be dominant if so?

Version 1:

Reviewer comments:

Reviewer #1

(Remarks to the Author)

The authors have responded to all the comments and adequately revised the manuscript. In particular, I think that the novelty of this study has been clear. Therefore, the revised manuscript is believed to be worthy of acceptance.

Reviewer #2

(Remarks to the Author)

The authors has addressed all of my concerns, and now I think the revised manuscript can be published in Nature Communications.

Reviewer #3

(Remarks to the Author)

Although I find that many parameters influencing the authors' results require further study before drawing solid conclusions, the authors have adequately addressed my previous concerns. The distinction between lateral and depth-dependent

inhomogeneities is now clearly articulated, and the discussion appropriately places their results in context with prior work, including that of Graves et al. Their response regarding the conditions under which the inhomogeneous mechanism may dominate is reasonable and well explained.

I am satisfied with the revisions and support the publication of the manuscript in its current form.

Reply to the referees – manuscript NCOMMS-25-20050-T

We would like to thank the referees for carefully reviewing our manuscript and providing constructive and generally very positive feedback. Please find our answers to the individual comments below in blue colour. Besides the revised manuscript, we also provide a red-lined version of the manuscript which includes all changes marked in colours. All line numbers and bibliography references mentioned in the following refer to the colour-lined version of the manuscript.

Referee #1:

Referee #1: The authors have investigated the mechanism of all-optical helicity-independent magnetization switching (AO-HIS) in GdCo multilayer structure using pump-probe technique. The space and temporal evolutions of magnetization have been measured through MOKE. They show that the AO-HIS is governed by the propagation of the domain-like region in the depth direction. It is commendable that non-uniform magnetization reversal was clearly observed. However, non-uniform reversal itself has already been reported (as ref. 37 in the manuscript) and thus, it cannot be said to be a novel phenomenon.

Response: We appreciate the referee's positive opinion on the clarity and unambiguity of our experimental results. However, we respectfully disagree with the statement that the observed non-uniform reversal is not a novel phenomenon and has already been reported.

We would like to emphasize that the non-uniform switching mechanism proposed by M. Beens et al., *Phys. Rev. B* 100, 220409 (2019) (Ref. 37, now Ref. 41) has so far been only a theoretical model, which is employed in spin dynamics simulations to explain the emergence of AO-HIS in Co/Gd bilayers. In fact, while they compare the dynamics of alloys and multilayers, they have so far investigated a propagation-based switching only in the case of bi- and multilayers, but not even considered this as a potential contribution to the AO-HIS of alloys.

We would like to emphasize two important points that demonstrate the originality and novelty of our work, setting it apart from that of Beens et al.: First, we provide direct experimental evidence of an inhomogeneous switching mechanism. Second, we demonstrate that this phenomenon is far more general and also occurs in amorphous alloys, not just in bilayers, which are a substantially different and special case. In bilayers, the initial nucleation of AO-HIS can only occur at the interface between the rare-earth and the transition metal, fully relying on a mechanism that propagates the switching into the transition metal layer; hence, the non-uniform reversal is imposed by the inherent structure of the bilayer. In case of amorphous alloys, especially if the magnetic film itself is homogeneous along the depth and only a few nanometres thin, such a strong influence of non-locally propagating AO-HIS

has not been taken into account so far.

We therefore believe that our findings fulfil the criteria of novelty and significance, especially due to the first experimental demonstration and the far-reaching implications for the design of magnetic heterostructures optimised for AO-HIS, but also, more generally, for ultrafast phase transitions within nanometre-thin layers.

Referee #1: In order to publish this manuscript in Nature Communications, I think that the following points need to be clarified.

1. Whether both up-to-down and down-to-up magnetization reversals proceed with the same mechanism is important. But in this manuscript, I can see only one of them.

Response: We must admit that we are not entirely sure whether we have understood the referee’s question correctly. In case that “up-to-down” and “down-to-up” refers to the switching propagating either from the top to the bottom of the GdCo layer or vice-versa, we would refer the referee to our reply to the next question (2.), where we discuss the effect of different cap and seed layers which could potentially reverse the direction.

If it is referring to the actual magnetisation direction, i.e., the polarity of the in-plane magnetisation in the GdCo layer, we thank the referee for pointing out that this aspect might not be clear enough in the manuscript.

First of all, we would like to note that our GdCo sample possesses an in-plane magnetic anisotropy with an easy-axis parallel to the surface, along which the magnetisation tends to align, and along which we constantly apply a magnetic field to restore the magnetisation after each pump-probe cycle. This is shown in Fig. 1a, but also explicitly mentioned in line 75.

However, there is no preferred direction along the easy-axis itself, i.e., the magnetic properties and dynamics of the sample are entirely independent of whether the magnetisation is initially aligned parallel or antiparallel to a particular direction along this easy-axis. Accordingly, magnetic hysteresis loops recorded along the easy-axis are perfectly symmetrical and do not change if the sample is rotated by 180° in the surface plane (compare Supplemental Material of J.-X. Lin et al., Phys. Rev. B 108, L220403 (2023)). For this reason, the induced dynamics are also totally independent of the initial magnetisation direction.

Additionally, as in all common pump-probe techniques, achieving magnetic contrast via magneto-optical Kerr effect (MOKE) or magnetic circular dichroism (MCD), the data is recorded using a difference scheme: relying on the direction-independent behaviour, the magnetic contrast is obtained by periodically flipping a saturating magnetic field applied along the easy-axis of the sample, alternating between opposite polarities of the initial saturation direction that is restored after each pump-probe cycle. The magnetisation transient is then obtained from the asymmetry, i.e., the normalized difference

between the two signals recorded for opposite saturation direction. This difference scheme is highly efficient, as it provides an intrinsic normalisation of the transient magneto-optical signal to the unpumped magnetisation of the sample (given by the difference signal at negative delays), mostly cancelling out any non-magnetic contributions and fluctuations of the photon flux of the source.

Therefore, the initial magnetisation direction along the easy-axis for the data presentation in the manuscript can be chosen arbitrarily, and it would be redundant to show also (completely symmetric) data for the opposite direction. For the revised version of the manuscript, we added a note on the direction-independent properties and dynamics along the in-plane easy axis to the Methods section (see lines 392–399):

The evaluated observable is the magnetic asymmetry, i.e., the normalised difference between the two spectra as a function of pump–probe delay (see Fig. 2). Employing such a difference scheme to obtain magnetic contrast is reasonable, as the studied $Gd_{25}Co_{75}$ sample possesses an in-plane magnetic anisotropy with an easy-axis parallel to the surface, along which the magnetisation has no preferred direction [21]. Thus, the magnetic properties of the sample and the induced dynamics are entirely independent of whether the magnetisation is initially aligned parallel or antiparallel to a particular direction along this easy-axis.

Referee #1: 2. The Pt cap layer plays an important role. What is the reversal mechanism when Pt is used as the bottom instead of the cap? In addition, how is the switching behavior affected when the cap layer is changed to a different material, especially one with low absorption?

Response: The referee is right that the choice of cap and seed layers can strongly influence the dynamics of the magnetic layer. For the GdCo switching sample studied in our current work, calculations of the 2.1 μm pump pulse absorption show that the deposited energy is largest in the Pt cap layer, which can therefore act as a source of heat and hot electrons that impinge the magnetic layer from the top. Although the presence of a Pt layer is not a mandatory requirement for switching in this type of systems, it explains why, in our case, the switching initially starts at the top of the GdCo layer.

In an earlier experiment (see M. Hennecke et al., *Phys. Rev. Research* **4**, L022062 (2022), now Ref. 20 in the manuscript), we investigated a GdFe sample in which the Pt was not used as a cap but as a seed layer. Although this was not a sample exhibiting AO-HIS, the data revealed the laser-induced demagnetization to be strongly enhanced at the bottom interface to the Pt seed layer; this could be attributed to the intriguing situation that the largest amount of pump energy was actually deposited in the Pt layer *below* the GdFe. We would therefore guess that reversing the layer stack of the switching sample, i.e., shifting the Pt from top to bottom, would cause the switching to start

at the bottom of the GdCo layer and propagate towards the top. We believe that this is one of the key findings of our work: it highlights the importance of careful heterostructure design, as the heterogeneous layers surrounding the magnetic film can easily change or introduce asymmetric excitation conditions that strongly affect the dynamics.

While without a solid theoretical framework, we cannot yet make robust predictions about how the dynamics are affected by using completely different cap and seed layer materials, we plan to continue with exactly this type of systematic study and investigate the effects of different heterostructure designs and material choices. We believe that this also opens a new route for tuning magnetic heterostructures for a desired functionality, e.g., achieving faster and more efficient switching. We also added this information to the discussion in the revised version of the manuscript (lines 212–219):

Note, that this observation is also in line with earlier depth-resolved studies on the ultrafast demagnetisation of a GdFe-based system, which have revealed that the strong absorption in Pt can even lead to an enhanced demagnetisation at the bottom of the magnetic film, when the Pt is used as a seed instead of a capping layer [20]. It further demonstrates how strongly the dynamics of the magnetic layer can be influenced by the heterostructure design, in particular the choice of the surrounding layers, which also opens a potential route for tuning, e.g., directionality and energy efficiency of the switching.

Referee #1: The followings are my questions and comments which will help to revise the manuscript.

1. Which sublattice magnetization is dominant at room temperature, Co or Gd? Why did you probe the Gd sublattice magnetization?

Response: We thank the referee for noticing that this information was indeed missing in the current version of the manuscript. All of our measurements were carried out at room temperature, which is *below* the ferrimagnetic compensation temperature of the studied Gd₂₅Co₇₅ alloy, thus, the Gd sublattice magnetisation is dominant. We agree that this is an important information, so we added it to the revised version of the manuscript (see lines 79–82):

*For the studied Gd₂₅Co₇₅ alloy, the ferrimagnetic compensation point is above room temperature, where all measurements were carried out. We enable an ultrafast and element-selective view on the **dominant** Gd sublattice magnetisation by femtosecond TMOKE spectroscopy in the soft-X-ray spectral range.*

Please note that further information about the sample system studied in our work can be found in J.-X. Lin et al., *Phys. Rev. B* **108**, L220403 (2023), which we cite as Ref. 21.

The choice for probing the Gd sublattice magnetization is mainly motivated by the more complex and pronounced spectral features of the Gd N_{5,4} resonance around ≈ 148 eV compared to the Co M_{3,2} resonance around ≈ 60 eV (as unfortunately, the Co L_{3,2} resonance at about 780 eV is still out of reach for

HHG-based light sources). Since the depth resolution results from a combination of varying penetration depth along the resonance with inter-layer reflection and interference effects, both the presence of more distinct spectral features as well as the shorter wavelength increase sensitivity and facilitate data analysis. However, for the future we plan to apply our method also to time-resolved measurements carried out at the Co $M_{3,2}$ resonance, as we agree that comparative measurements of the Co sublattice magnetisation will complement the picture of boundary-driven ultrafast magnetisation reversal.

Referee #1: 2. In Fig. 3(b), magnetization depth profiles are shown. What is the origin of the CoGd magnetization decrease near the Cu/CoGd and CoGd/Ta interfaces for unpumped state (-1.0 ps)?

Response: We agree that this is an observation that indeed deserves explanation, which is why we already included a note on the non-uniform magnetisation distribution in the unpumped state (see lines 459–466 of the methods section describing the magnetic scattering simulations).

This decrease which is both present at negative delays (-1.0 ps) as well as in the completely static measurements (see Extended Data Figure 3) can be attributed to effects like inter-layer diffusion, which are known from the literature (see, e.g., M. Kowalewski et al., *J. Appl. Phys.* **87**, 5732-5734 (2000) and F. Hellman et al., *Rev. Mod. Phys.* **89**, 025006 (2017) to reduce the magnetisation at the interfaces with other non-magnetic materials. Additionally, a slight effect of static heating induced by the pump laser can be observed, as this decrease gets even more pronounced for higher excitation fluence.

We now feel that this information may be buried too deep in the methods section and have therefore moved it to the main text where the data in Fig. 3 is discussed (see lines 132–139):

Already before the pump pulse excites the sample (-1.0 ps), the fit converges for a non-uniform distribution of magnetisation within the $Gd_{25}Co_{75}$ layer, slightly decreasing towards the neighbouring Cu and Ta layers. This observation can be attributed to effects like inter-layer diffusion, reducing the magnetisation at the interfaces with other non-magnetic materials [29, 30]. Furthermore, static heating induced by the repetitive absorption of the pump radiation also affects the magnetisation of the unpumped state, which is particularly apparent by the fluence-dependent decrease towards the Cu interface.

Referee #2:

Referee #2: The study by Hennecke et al. provides a transformative perspective on the mechanisms underlying all-optical helicity-independent switching (AO-HIS) in ferrimagnetic GdCo alloys, revealing that magnetization reversal is driven by a transient domain boundary propagating along the depth of the magnetic layer. By combining time-resolved soft-X-ray spectroscopy (TMOKE) with magnetic scattering simulations, the authors demonstrate that the process is spatially inhomogeneous, challenging the prevailing view of AO-HIS as a purely local, thermally driven process. The depth-resolved dynamics show a bi-domain magnetization state emerging post-excitation, with a boundary moving at ≈ 2000 m/s, a speed comparable to current-induced domain wall velocities in ferrimagnets. This propagation is not explained by thermal diffusion alone, as supported by two-temperature model (2TM) simulations, and instead implicates non-local mechanisms such as spin-Seebeck effects or exchange interactions.

The experimental design is rigorous, with structural parameters validated via hard X-ray reflectometry and sensitivity analyses confirming the necessity of inhomogeneous models over homogeneous switching assumptions. Fluence-dependent measurements reveal a threshold behavior (5.5 mJ/cm²), below which partial reversal relaxes and above which the boundary propagates to completion. These findings are bolstered by spectral asymmetry data and late-time magnetization profiles, ensuring robust validation of the proposed mechanism. The study's strengths lie in its innovative methodology, which provides the first direct observation of depth-resolved AO-HIS dynamics, and its quantitative analysis linking domain wall velocities to established spintronic phenomena.

While the work is focused on GdCo alloys, the implications extend to broader research on ultrafast phase transitions and nanoscale magnetism. A minor limitation is the unresolved theoretical specifics of the boundary propagation mechanism, which the authors acknowledge and suggest for future exploration. The presentation is clear, with well-constructed figures and supplementary data (e.g., Extended Data) offering comprehensive validation. Although the textual description suffices to convey key insights, the supplementary video enhances understanding of domain dynamics.

The significance of the study lies in updating the understanding of AO-HIS, emphasizing spatially inhomogeneous dynamics critical for ultrafast switching. This work not only advances fundamental knowledge but informs the design of energy-efficient spintronic devices. Recommended revisions include briefly addressing the potential generality of the mechanism to other materials and clarifying theoretical open questions. Overall, the manuscript meets the criteria of Nature Communications for technical rigor, novelty, and broad relevance, warranting publication after minor revisions to enhance contextual discussion

and mechanistic clarity.

Response: We are delighted with the very positive assessment of our work and thank the referee for the strong recommendation for publication in *Nature Communications*.

Speaking of generality, we fully agree with the referee that our results point at far-reaching implications that extend beyond AO-HIS in GdCo or comparable ferrimagnetic alloys. While AO-HIS is a fascinating phenomenon by itself, the direct evidence that even on a few-picosecond time scales and a few nanometre length scales, the ultrafast switching of an order parameter can be driven by the propagation of a highly non-equilibrium boundary, suggests that similar inhomogeneities must also be considered for (non-magnetic) switching scenarios and phase transitions in other types of materials. This becomes particularly relevant for any non-symmetric boundary and/or initial conditions, as they are inherent to nearly every laser-driven pump-probe experiment and for materials which are embedded in heterostructures with built-in heterogeneities along the depth of the sample.

In this respect, it is advantageous that our method of obtaining depth resolution is generally not limited to the investigation of *magnetic* effects or transitions. It merely requires a spectroscopic observable which can clearly be related to the phase transition under investigation, i.e., the knowledge about how the initial and final state is identified in the spectrum. We therefore believe that our work will both stimulate future investigations on a broad range of ultrafast phase transitions as well as provide a versatile method to study them in depth.

With respect to the clarification of the exact mechanism driving the inhomogeneous AO-HIS in the studied system, the referee is right that the physics behind the propagation of the transient domain boundary remains an open question that urges for further investigation. Therefore, we plan combined experimental and theoretical efforts to tackle the remaining question and obtain insights on the fundamental microscopic mechanism driving the observed dynamics. This includes systematic studies of different heterostructure designs (e.g., variation of alloy composition, variation of interfaces including different cap and seed layers) and excitation conditions (e.g., at different temperatures), as well as atomistic spin dynamics simulations which theoretically model the inhomogeneous AO-HIS in our system.

Following the referees suggestion, we added a brief note on the generality of our results and method to the revised version of the manuscript, elaborating also on the outlook regarding clarification of the switching mechanism (see lines 297–304, 308–312 and 323–331):

To this end, the variation of the stacking sequence, the capping and seed layer materials, as well as the composition and thickness of the magnetic film will enable direct control over the layer-dependent absorption and, accordingly, the formation, size, and direction of excitation gradients and related transport

phenomena. Hence, a systematic investigation of AO-HIS for well-considered heterostructure designs will shed light on the conditions under which non-local, propagation-based switching can be observed, as well as on the actual mechanism driving the propagation of the transient boundary.

...

Furthermore, modelling the observed inhomogeneous dynamics, e.g., employing atomistic spin dynamics simulations, could help to identify the key ingredients for the occurrence of either local or non-local switching processes and also to transfer the obtained insights to switching phenomena in other types of magnetic materials.

...

This becomes particularly relevant when the active layer is embedded in a heterostructure, as the surrounding layers can easily cause an inherent inhomogeneity along the depth, leading to asymmetric excitation conditions that facilitate the formation of inhomogeneous dynamics or even determine their direction. We expect these findings to be relevant for phase transitions driven via non-equilibrium conditions in general. In this context, we would like to conclude by noting that our method for obtaining depth resolution is merely based on a spectroscopic observable and thus not limited to the investigation of magnetic effects. Therefore, we anticipate its application also for a wider range of non-magnetic phenomena.

Referee #3:

Referee #3: This study explores ultrafast all-optical helicity-independent magnetisation switching (AO-HIS) in sub-10 nanometre GdCo alloy films. Using femtosecond X-Ray transverse magneto-optical Kerr (TMOKE) spectroscopy at the Gd resonance, the authors track the absorption depth-resolved evolution of magnetisation. The results reveal that switching is not uniform across the film; rather, it initiates near the surface and propagates inward via a transient domain boundary. This non-local switching process is driven by the platinum capping layer, which absorbs the infrared pump energy and transfers heat and hot electrons into the underlying GdCo, triggering magnetisation reversal. These findings highlight the critical role of interface-driven dynamics and challenge the conventional view of AO-HIS as a purely local phenomenon. Although questions regarding non-local dynamics in GdFeCo have been raised previously—most notably by Graves et al. (Nature Materials, 2013), who showed they can originate from the sub-10 nm patches of REs and TMs—this work provides a compelling and technically advanced demonstration using state-of-the-art soft X-ray spectroscopy, supported by detailed modelling.

Response: We would like to thank the referee for this encouraging feedback and also for pointing to the earlier work of C. E. Graves et al. on non-local *lateral* dynamics during HI-AOS, which we will discuss below (see second last issue by Referee #3).

Referee #3: I have a few questions for the authors:

The fitting procedure appears to involve a large number of parameters. How unique and robust are the spatial magnetisation profiles derived from the simulations?

Response: The referee is of course right that the magnetic scattering simulations used to describe the data depend on a large number of parameters. Besides the magnetisation profile, these include the atomic and magnetic form factors as well as the structural properties (thicknesses, densities, roughnesses) of the individual layers. However, as detailed in the Methods section, the number of parameters which are actually *free* and used for fitting the time-resolved data is greatly reduced.

In particular, the form factors and the structural parameters are independently determined with high accuracy via reference and hard-X-ray reflectometry measurements (see the corresponding Methods section) and kept constant while fitting the magnetisation profiles. Evaluation of the transient sum signal, measuring the non-magnetic contribution to the dynamics, confirms that this is indeed justified (compare Extended Data Figure 7).

The very good agreement of the simulations with the static TMOKE data over the entire range of incidence angles ϑ (see the asymmetry map in Extended

Data Figure 3) serves as a meaningful benchmark for the accurate description of our system.

At the end, the only free parameter that is fitted to the time-resolved data is the magnetisation profile. As described in the Methods section, we model the profile by a second-order polynomial function, which is the simplest analytical model allowing a consistent description of both static and time-resolved data. Thus, the number of parameters used for fitting the asymmetry spectra at each pump-probe delay is reduced to the three coefficients of the second-order polynomial function.

Regarding the significance and uniqueness of the fitted magnetisation profiles, we would like to refer the referee to Extended Data Figure 6, which analyses the residual of the fits for artificial slopes and offsets added to the best fits of the magnetisation profiles. This analysis highlights the sensitivity and robustness of our method, showing that for deviations larger than 2%, the simulations would already be inconsistent with the experimental data.

Referee #3: What would happen if a different capping material were used instead of platinum? How critical is Pt for the observed dynamics? What if gold is used instead?

Response: To answer this question, we would first like to refer the referee to the very similar question raised by Referee #1. The results of our previous (see M. Hennecke et al., *Phys. Rev. Research* 4, L022062 (2022), Ref. 20 in the manuscript) and current work clearly show that the choice and geometry of the surrounding cap and seed layers can strongly affect the dynamics close to the interfaces. In particular, the high absorption of the 2.1 μm pump pulses in the Pt layer leads to the laser-induced demagnetisation and switching to be clearly enhanced close to the respective interface.

We agree that investigating the influence of different types of materials and interfaces on the switching dynamics is a very interesting follow-up, which is certainly on our agenda.

Referee #3: To what extent is the observed depth profile governed by interfacial effects versus the overall thickness of the GdCo layer? Could this be systematically studied, either experimentally or through simulations?

Response: On the experimental side, we plan to continue by varying the GdCo layer thickness and the interfaces (e.g., by exchanging cap and seed layers, and using different materials), as well as the pump laser wavelength.

Additionally, in collaboration with theory, we started first efforts to model the observed switching dynamics in alloys using atomistic spin dynamics simulations. Both approaches in conjunction will allow us to systematically investigate the switching dynamics for different interfaces and alloys, hopefully obtaining more insights on the microscopic processes behind the observed

dynamics.

As this might be an interesting question also for the broad audience, we added this as an outlook to the revised manuscript (see lines 297–312):

To this end, the variation of the stacking sequence, the capping and seed layer materials, as well as the composition and thickness of the magnetic film will enable direct control over the layer-dependent absorption and, accordingly, the formation, size, and direction of excitation gradients and related transport phenomena. Hence, a systematic investigation of AO-HIS for well-considered heterostructure designs will shed light on the conditions under which non-local, propagation-based switching can be observed, as well as on the actual mechanism driving the propagation of the transient boundary.

...

Furthermore, modelling the observed inhomogeneous dynamics, e.g., employing atomistic spin dynamics simulations, could help to identify the key ingredients for the occurrence of either local or non-local switching processes and also to transfer the obtained insights to switching phenomena in other types of magnetic materials.

Referee #3: Do the results support or suggest the possibility that hot electron currents can be used to induce magnetisation switching in GdCo?

Response: This raises a very interesting point. Indeed it has been shown already some time ago that AO-HIS in GdFeCo samples can be driven by hot electron pulses, e.g., by R. B. Wilson et al., *Phys. Rev. B* 95, 180409(R) (2017) or Y. Xu et al., *Adv. Mater.* 29, 1703474 (2017). Here, one can expect that the hot electrons mainly excite the interface region of the GdFeCo sample due to the very small mean free path of electrons in GdFeCo and that the switching is likely to exhibit a depth dependence. Again, this is a very interesting scenario, which we have already started to work on. In the manuscript, we now also explicitly mention the possibility of triggering switching solely by hot electron pulses (see lines 209–212):

Such hot electron pulses are known to induce ultrafast switching in GdFeCo-based heterostructures, even when the highly conductive spacer layer is thick enough to prevent direct optical excitation of the magnetic film [32, 33].

Referee #3: The study reveals spatial inhomogeneity along the film depth. In Graves et al., lateral inhomogeneity with characteristic lengths of ≈ 10 nm was also observed. Can the authors comment on the significance of this comparison? Is there any link between lateral and depth inhomogeneities in these systems?

Response: We thank the referee again for pointing out the seminal work by C. E. Graves et al. and for asking how it relates to our findings. Of course, we have been asking ourselves the same question and found some major differences independent of the spatial dimension, along which the inhomogeneous dynamics occur:

(i) in Graves et al., the GdFe sample exhibits an intrinsic structural inhomogeneity, namely lateral patches of Gd- and Fe-rich regions. Such structural heterogeneity is not observable along the depth of our GdCo sample, as TEM measurements have shown.

(ii) they observe spin currents between the inhomogeneities that arise on the early time scales (even though the accumulated difference is long-lived).

In our data, we do not observe any indications for spin currents, instead, the inhomogeneous dynamics start to evolve only after thermalisation. Despite these differences between the two effects, it is of course possible that they occur simultaneously, but are not necessarily related or affect each other, and that they might be strongly sample-dependent (e.g., due to lateral inhomogeneities, interfaces, ...). While we average over any lateral inhomogeneity, Graves et al. have averaged over any depth-dependent inhomogeneity, and it would require a combined experimental effort to probe the inhomogeneous dynamics along both dimensions simultaneously. Motivated by this discussion, the referee has stimulated, we have added a paragraph to the introduction describing how our work differs from that of Graves et al. (see lines 55–63):

While it has already been shown that femtosecond laser excitation can induce non-local dynamics along the lateral dimension, e.g., by an ultrafast spin transfer between chemical inhomogeneities within an amorphous GdFeCo alloy [19], another significant inhomogeneity exists along the depth of a typical nanolayer system, in which the magnetic film is surrounded by distinct capping and seed layers. This inherent heterogeneity, although not originating from any structural or chemical inhomogeneities within the magnetic layer itself, give rise to asymmetric excitation conditions which potentially affect the optically excited dynamics [20].

Referee #3: Does it happen that the suggested inhomogeneous mechanism dominates the thermally-induced uniform switching in GdFeCo? When can it be dominant if so?

Response: By inspection of Fig. 5a, one can appreciate that the propagating AO-HIS dominates for a fluence slightly above the switching threshold, as only a small fraction of the GdCo layer is driven into the direct, purely thermal switching regime. In general, we expect the inhomogeneous mechanism to play a role whenever an inhomogeneous excitation profile is present. Due to the short penetration depth of optical light in metals and the typical design of thin magnetic films with distinct seed and capping layers, homogeneous excitation is difficult to achieve. However, the fluence dependence on early time scales (see top panel of Fig. 5a) suggests that one could still increase the fluence up to a point where the entire magnetic layer is thermally driven into the switching regime, which could enable full switching without a second, inhomogeneous process. Unfortunately, we could not verify this with the studied sample, as the increased heat load prevented us to use fluences above 6.0 mJ/cm^2 without

inducing damage. Our data strongly suggest, though, that the dominance of homogeneous or inhomogeneous processes can be controlled by the design of the heterostructure, the excitation conditions, and possibly also the equilibrium temperature. Actually, as soon as the mechanism can be described by theory, it should be possible to specifically adjust these parameters in such a way that either one or the other mechanism dominates. This is particularly appealing, since the ability of non-local switching observed in our experiment could further extend the usually narrow parameter range in which AO-HIS can be observed, making it highly desirable from a technological point of view.